# Review on Perovskite Semiconductor Field–Effect Transistors and Their Applications

**DOI:** 10.3390/nano12142396

**Published:** 2022-07-13

**Authors:** Gnanasampanthan Abiram, Murugathas Thanihaichelvan, Punniamoorthy Ravirajan, Dhayalan Velauthapillai

**Affiliations:** 1Department of Physics, University of Jaffna, Jaffna 40 000, Sri Lanka; vgnabi28@gmail.com (G.A.); pravirajan@univ.jfn.ac.lk (P.R.); 2Department of Computer Science, Electrical Engineering and Mathematical Sciences, Western Norway University of Applied Sciences, Inndalsveien 28, 5063 Bergen, Norway

**Keywords:** perovskite, field effect transistor, photo detector, light-emitting FET, mobility

## Abstract

Perovskite materials are considered as the most alluring successor to the conventional semiconductor materials to fabricate solar cells, light emitting diodes and electronic displays. However, the use of the perovskite semiconductors as a channel material in field effect transistors (FET) are much lower than expected due to the poor performance of the devices. Despite low attention, the perovskite FETs are used in widespread applications on account of their unique opto-electrical properties. This review focuses on the previous works on perovskite FETs which are summarized into tables based on their structures and electrical properties. Further, this review focuses on the applications of perovskite FETs in photodetectors, phototransistors, light emitting FETs and memory devices. Moreover, this review highlights the challenges faced by the perovskite FETs to meet the current standards along with the future directions of these FETs. Overall, the review summarizes all the available information on existing perovskite FET works and their applications reported so far.

## 1. Introduction

Perovskite has been the most conspicuous material in the field of photovoltaics for more than a decade, as it shows a huge improvement in the power conversion efficiency [1]. The outstanding intrinsic optoelectronic properties of perovskites, such as tunable band gap, relatively high career mobility, diffusion length, high absorption co-efficient and photoluminescence quantum yield, along with the advantages of better feasibility, simpler processing ability and more flexibility than the conventional semiconductors during the fabrication of devices, make it one of the most highly researched materials in the second decade of the 21st century [2,3,4]. These advantages of perovskites permit numerous electrical, optical and magnetic applications in the electronics industry [5,6,7,8]. Field effect transistors (FETs) are the most important electronic components for current consumer electronic devices. A FET consists of a semiconducting channel with three terminals: the source, drain and gate. A thin-film FET is a FET with a thin-film semiconductor material as the channel. The basic working principle of a FET is that the channel resistance between the drain and source terminals is controlled by the gate terminal [9]. In FETs, the gate terminal is electrically isolated from the channel, and the channel resistance can be controlled capacitively by an electric field.

A FET with a semiconducting perovskite channel is often referred as a perovskite FET. Despite their excellent electro-optical properties, perovskites have not been fully exploited as a semiconductor material in field effect transistor (FET) applications when compared to solar photovoltaic applications [10]. The hybrid perovskite material MAPbI_3_ (CH_3_NH_3_PbI_3_) (MA—Methyl ammonium) is the most studied perovskite material for applications in FETs [11]. Although MAPbI_3_ was introduced as an absorber material in PV applications in 2009 [12], the first ever MAPbI_3_ perovskite channel-based FET with the same material was reported only in 2015 [13]. 

Figure 1 shows various types of perovskite materials, which are used to perform different functions in thin-film FETs. In 1997, the perovskite material La_0.7_Ca_0.3_MnO_3_ (LCMO) was used as a semi-conducting channel, and another perovskite structured PbZr_0.2_TiO_0.8_O_3_ (PZT) was used as a ferroelectric layer in a thin-film FET [14]. However, the introduction of a hybrid double perovskite (C_6_H_5_C2H_4_NH_3_)_2_SnI_4_ ((C_6_H_5_C_2_H_4_NH_3_)—phenethyl ammonium (PEA)) in the year of 1999 laid the foundation for the research works on semiconducting perovskite thin-film channel FETs [15]. The perovskite (C_6_H_5_C_2_H_4_NH_3_)_2_SnI_4_ was used as a semiconductor material in thin-film FETs by a group at the IBM T. J. Watson research center [15,16] by using different synthetic processes [15]. The highest proportion of perovskite materials have therefore been used as the channel materials in thin-film FETs, as shown in Figure 1. 

In most of the reported perovskite FET works, the FET structure is used to study the charge carrier dynamics and mobility of the perovskite materials [17]. However, in recent past, the perovskite FETs have attracted global attention due to the widespread applications of these perovskite FETs in light detectors [18], photo FETs [19], light-emitting FETs (LEFET) [20], and static and dynamic memory devices [14]. Yet the application of perovskite materials as the channel in FETs needs many further studies to become a potential contender in the commercial market.

Quite a few reviews of the research works on perovskite FETs have been carried out based on many different points of view, such as material types, fabrication technologies, and particular applications [21,22,23,24,25,26]. The studies on the fabrication and performance of the metal halide perovskite FETs were reviewed in [24]. Wu et al. published a review article on the application of perovskite materials in FETs [21]. Studies of perovskite-based phototransistors and photo detectors [27,28], inorganic and hybrid perovskite-based lasers [29], and metal halide perovskite-based radiation detectors [30] were reviewed by focusing on the particular application of perovskite materials. The application of various perovskite materials used for the fabrication of FETs was reviewed by Wu T et al. [21]. Works on perovskite-based FETs were marginally analyzed in the reviews of all the optoelectronic and photovoltaic applications of perovskite materials [21,26]. The works on the fabrication of perovskite thin-film FETs were presented while mainly conducting reviews on perovskite thin-film fabrication techniques [25,31]. Cai, W et al. reported a mini review on double-perovskite-based FETs [32]. Recently, Paulus, F et al. reported an elaborated review study on perovskite-based FETs and their applications [11].

This study reviews the device performance of thin-film FETs with various perovskite semiconductor materials functioning as the active channel and perovskite FET applications. Tables with reported perovskite material channel based thin film FETs and perovskite thin film FETs’ applications are classified into different categories and the functional principles are compared and analyzed. The perovskite thin film FETs in the literature are categorized in two timelines in this review. A rapid increase in the citations of perovskite based studies was observed after 2012 [33]. A similar trend is obvious with the number of perovskite FET studies as only a few perovskite materials were used in the thin film FETs before 2012. We did not come across any reports on perovskite FETs between 2012 and 2015. However, after 2015, a number of perovskite materials have been extensively studied as the semiconducting channel materials in thin-film FET studies and the applications of perovskite FET. We categorized the perovskite FETs based on the three types of perovskite semiconducting materials (Hybrid, all-inorganic and Double and triple perovskites). The main focus of this review is the applications of the perovskite thin film FETs, which are analyzed under five different categories: (1) FET platform as a tool for evaluating the mobility, (2) Photo FETs, (3) Photo detectors, (4) Light-emitting field effect transistors LEFETS and (5) Ferroelectric random access memories (RAM). Finally, we outlined the major challenges and discussed the future directions of the perovskite FET research works.

## 2. Fundamentals of Semiconducting Perovskite Semiconductor Materials

The structure of perovskites is considered as octahedral with the generic chemical formula of ABX_3_. The term perovskite is used to denote the materials existing in the crystal structure of ABX_3_. Perovskites are bonded in the octahedral shape of corner sharing structure, which forms a framework of 3D crystal structure as shown in Figure 2.

Perovskites are extensively used in different dimensions, such as 0D nanoparticles [34], 1D nanowires (NW) [35], 2D nanosheets [36] and single crystals (SC) or thin films [37,38] in optoelectronic device applications ranging from thin-film light-emitting devices, solar cells and solid-state lasers to thin film FETs. Perovskite semiconductor materials are synthesized by varying the substitutions for each component with different molecular stoichiometry [39]. Different cations and anions, which are shown in Figure 3, incorporated with the compositions of the perovskites, dictate the performance of the devices and the fabrication processes as well [1].

The certain semiconducting properties such as structure, stability and mobility of the resultant perovskites, composed from the ions stated in Figure 2 depend on the attributes of respective formulating ions. So, the perovskites are classified according to the compositions of ions as single hybrid perovskite, all inorganic perovskite and the double and triple perovskites. For instance, a double perovskite and a triple perovskite can be formulated by the different ions in Figure 3 with multiple stoichiometric compositions.

In a perovskite of ABX_3_, as the cation of A is surrounded by the eight anionic ${\mathrm{BX}}_{6}^{-}$ in three-dimensional cubic shape as shown in the Figure 2. The stability of the perovskite materials is decided by the chemical components which formulate the material. Goldschmidt’s tolerance factor t is used as a formula to predict the stability of the perovskite structure.
$$t=\frac{R_{A}+R_{X}}{\sqrt{2(}R_{B}+R_{X})}$$
where $R_{A},\;R_{B}\;\mathrm{and}\;R_{X}$ are the ionic radii of cation *A*, the divalent metal ion and the halogen ion, respectively. At room temperature, Goldschmidt’s tolerance factor for a stable perovskite cubic structure ranges from 0.89 to 1 [40,41].

### 2.1. Hybrid Perovskites

Hybrid perovskite is formulated by substituting organic cations such as methyl ammonium (CH_3_NH_3_^+^), ethyl ammonium (CH_3_CH_2_NH_3_^+^), 4-fluorophenyl ethyl ammonium 4-FPEA, formamidinium (FA^+^), 3- amino propyl and other alkyl ammonium ions (RNH_3_^+^), etc., in place of A of ABX_3_ with the metallic (inorganic) cation of B and the halide or oxide anion of X [31,39,42]. Subsequently, these are classified as hybrid (organic–inorganic) perovskites. The metal halide hybrid perovskites are the most exciting materials in optoelectronic applications, as they have shown an excellent spectroscopic development [43,44].

In hybrid perovskites, the organic cation is larger than the metallic ions and has the ability to rearrange the crystal structure as the organic part is bonded by the weak van der Waal interactions and hydrogen bonding [34,35]. So, the symmetry of the molecule is increased in the hybrid perovskites more than the other types. As a result, the optoelectronic properties of the hybrid perovskites can be fine-tuned by varying the chemical structure [45]. The hybrid halide perovskites can be synthesized using simple and easy processes [46,47]. The instability of these hybrid materials in the exposure to the moisture, heat and the light in the environment is a major challenge during the fabrication [48] and operations of the devices [49,50], which keeps improving with further studies. However, the limitation in the value of the charge carrier mobility with organic cation barriers [51], as it does not vary with the different halogen anions, is considered as another concern to be overcome in the commercialization of devices based on hybrid halide perovskites [17,52].

### 2.2. All Inorganic Perovskites

The halide and oxide perovskites, which are formulated with monovalent or divalent inorganic A site cations of the ABX_3_ prototype, are known as all inorganic perovskites [53]. For instance, CsPbBr_3_, SrTiO_3_ or KTaO_3_ are generally categorized as all inorganic perovskites. With the presence of metallic ions, the high carrier mobility along with the narrow emission peak of these types of perovskites is considered as an advantage in optoelectronic applications [54]. Hence, the device performance using all inorganic perovskite semiconducting materials is massively improved compared to the other types of perovskites [55,56]. Therefore, all inorganic perovskites are being used in multiple device applications in different structural forms [57]. However, the brittleness and the very low colloidal stability of these materials are considered as a major challenge during the fabrication of the devices using all inorganic perovskites [58].

### 2.3. Double and Triple Perovskites

The double and triple perovskites have the generic formula of A_2_M(I)M(III)X_6_, A_2_M(IV)X_6_ or A_n+1_B_n_X_3n+1_, which were derived from the ABX_3_ prototype [59]. The double and triple perovskites came into the fold a little later than the single hybrid perovskites in the optoelectronic applications as the charge recombination occurs as a result of the high exciton-binding energy rather than the charge separation [60]. However, these types of perovskites are being synthesized to fabricate many different optoelectronic devices [61] due to their improved stability to temperature, light and moisture than the single hybrid halide and all inorganic perovskites. The structure-related optoelectronic properties are the main factors to be considered in the fabrication for device applications [62]. However, the ion migration hindered by the incorporation of the layers in these perovskites causes lower charge carrier mobilities in the device performances [63]. The stability of these types of perovskites was found to be better than the other two types due to their building blocks [64,65].

## 3. Perovskite FETs

### 3.1. Perovskite Thin Film FET Device Structures

Thin film FET consists of a thin film semiconductor material as the channel. In perovskite thin film FETs, the channel is generally a polycrystalline thin film made-up of perovskite structured semiconductors. It also consists of two metal electrodes of source S and drain D with Gate electrode of G which is located close to the channel layer. The gate, electrodes and semiconductors can be placed in different positions hence four different device configurations bottom gate top contact (BGTC), bottom gate bottom contact (BGTC) top gate bottom contact (TGBC) top gate top contact (TGTC)) to fulfill the different functional purposes as shown in Figure 4.

The position of injecting electrodes and the difference in certain interface contacts from one structure to another may affect the performance and the behavior of the FET device [66]. These device structures are classified into two types: coplanar configurations, where the planes of the source and drain are offsets to the plane of the semiconductor, and staggered configurations, where the source and drain electrodes and the semiconductor layer are on the same plane [67]. Most of the thin-film FETs were fabricated in the orientation of a bottom gate top contact (BGTC) device, where the metal electrodes of the source and drain are placed on top of the semi-conducting channel of the perovskite, as shown in Figure 4a. The bottom gate bottom contact (BGBC)-oriented device was fabricated by placing the metal electrodes of the source and drain in between the perovskite channel and the dielectric layer, coated on top of the bottom gate as shown in Figure 4b [68]. The perovskite photo-detecting thin-film FETs were fabricated in two different device structures: bottom gate top contact and bottom gate bottom contact [18]. The active light-detecting layer was attached on the top and bottom of the two different structured photo-detecting thin-film FETs. A top gate bottom contact (TGBC) thin-film FET was fabricated by placing a gate electrode on top of perovskite semiconducting channel, where the device was fabricated based on the substrate, as shown in Figure 4c, in search of constructing a hysteresis-free device [69] in the fabrication and characterization of ferroelectric gate FET [70]. In addition, the all inorganic BaSnO_3_ perovskites was tested as a channel material in a top gate top contact (TGTC) device in [71].

### 3.2. Electrical Properties of FETs

A FET can be electrically characterized in two ways: (1) output and (2) transfer characteristics. Among these, the transfer characteristics of the FET can expose its switching characteristics [72]. The transfer characteristic of a FET is the variation in the drain source current I_ds_ under constant drain source voltage *V_ds_* with varying gate voltage *V_g_*. Figure 5a,b show the p-channel and n-channel transfer characteristic curves respectively of a triple cation perovskite channel based thin film FET. Generally, the source electrode is considered as the charge-injecting one, so it is always more negative than the positive gate voltage to obtain the electron injection and always more positive than the negative gate voltage to obtain the hole injection. Most of the reported perovskite FETs are ambipolar FETs, which show both n type and p type characteristics at different gate biases [73,74]. The output characteristic curves of the p-channel and n-channels are given in Figure 5c,d, respectively [75].

#### 3.2.1. On-Off Ratio

The on-off ratio is an important measure of a FET which indicates the quality of the switching. This is a numerical ratio between the on and off-stage currents through the channel as shown in Figure 6. High on-off ratios indicate that the on currents of the FET are much higher than that of the off current and hence the FET is highly switchable. The off current in ambipolar FETs can be defined as the current when the carrier inversion occurs, thus the minimum current during the transfer sweep.

#### 3.2.2. Threshold Voltage

The threshold voltage V_T_ of a FET can be defined as the voltage at which the FET turns to the off state from the on state [72,77] as shown in Figure 6. In an ambipolar FET the threshold voltage will be the voltage at which the current I_ds_ minimum occurs. In other words, it is the voltage at which the carrier inversion occurs. The region between the turn on voltage and the threshold voltage is known as the sub threshold regime in the transfer characteristics curve and the reciprocal slope of the log (I_ds_) − V_g_ curve gives the sub threshold swing, which reflects the speed of switching ability of the device.

#### 3.2.3. Mobility Calculations in FETs

The most important intrinsic electrical properties of a semiconductor are its charge carrier mobility and diffusion length [51]. A good proportion of the perovskite-based thin film FET studies were carried out to measure the mobility of the semiconducting materials [78]. In a perovskite thin-film FET, the field effect mobility μ is considered as the average drifting velocity of the charge carriers in the active semiconducting layer with an applied electric field [79]. The efficiency of the FET devices is increased with the charge carrier mobility in the channel, as the drain source current primarily depends on it [79]. The calculation of charge carrier mobility in a semiconducting layer of a perovskite FET is carried out based on the transfer characteristics curve with two important assumptions: (a) mobility does not depend on carrier density, and (b) the transverse gate electric field is much greater than the applied source and drain voltage [80].

With the low applied V_ds_ values, $\left(\left|V_{g}-{\;V}_{T}\right|\gg V_{\mathrm{ds}}\right)$ hence the I_ds_ increases linearly in the transfer characteristics curves [81]. In this region, the drift velocity of the charge carriers in the perovskite channel is concerned as the linear mobility $\mu_{\mathrm{lin}}$ which is given by the equation of
$$I_{\mathrm{ds},\mathrm{lin}}=\frac{W}{L}C_{i}{\;\mu }_{\mathrm{lin}}(\left(V_{g}-V_{T}\right)V_{\mathrm{ds}}-\frac{1}{2}V_{\mathrm{ds}}{}^{2})$$

And from this the $\mu_{\mathrm{lin}}$ is extracted as
$$\mu_{\mathrm{lin}}=(\frac{{\mathrm{dI}}_{\mathrm{ds},\;\mathrm{lin}}}{{\mathrm{dV}}_{g}})\frac{L}{V_{\mathrm{ds}}w}C_{i}$$
where W and L stand for channel width and length respectively, $C_{i}$ is the gate capacitance per unit area. $I_{\mathrm{ds}}\;$ is the drain source current of the device with the applied gate source electric field of $V_{g}$. To find out the value of $\left(\frac{{\mathrm{dI}}_{\mathrm{ds},\;\mathrm{lin}}}{{\mathrm{dV}}_{g}}\right)$, the gradient of the linear regime transfer characteristics curve is calculated by the gradual channel approximation in the region of sub threshold swing [81] as shown in in Figure 6.

When $\left|V_{g}-V_{T}\right|\ll \left|V_{\mathrm{ds}}\right|$, the source drain current remains constant with the applied source drain voltage for the applied gate voltage. The drain current of I_ds_ in the saturation regime was given with the relation to the charge carrier mobility $\mu_{\mathrm{sat}}$ and applied gate-source voltage V_gs_ by the following equation [79]:$$I_{\mathrm{ds},\mathrm{sat}}=\frac{{\mathrm{WC}}_{i}}{2L}{\;\mu }_{\mathrm{sat}}{\left(V_{g}-V_{T}\right)}^{2}$$

As a derivation, most of the time, the field-effect mobility of linear active regime in a perovskite-based thin-film FET was calculated by the basic equation of [82]:$$\mu_{\mathrm{sat}}={\left(\frac{{\mathrm{dI}}_{\mathrm{ds}}{}^{1/2}}{{\mathrm{dV}}_{\mathrm{gs}}}\right)}^{2}\frac{2L}{{\mathrm{wC}}_{i}}$$

From the gradient of the blue curve in the Figure 6, the value of ${\left(\frac{{\mathrm{dI}}_{\mathrm{ds}}{}^{1/2}}{{\mathrm{dV}}_{\mathrm{gs}}}\right)}^{2}$ is obtained, w and L are the channel width and length, respectively, and $C_{i}$ is the gate capacitance per unit area [83].

### 3.3. Early Works (Before 2012) on Perovskite FETs

Based on the literature, the first research paper on perovskite FET was published in 1999 using a layered bulk (C_6_H_5_C_2_H_4_NH_3_)_2_ SnI_4_ thin film as the semiconducting channel with Pd as source and drain electrodes from the IBM T. J. Watson Research Center [15]. The same research group reported (C_6_H_4_FC_2_H_4_NH_3_)_2_SnI_4_ perovskite thin-film FETs in 2001 [16] and (phenethyl ammonium)_2_SnI_4_ in 2002 [84]. A group in Kyushu University of Japan continued the work based on (PEA)_2_SnI_4_ channel FETs by using different fabrication methods of deposition techniques in 2003 and 2004 [85,86]. In 2006, the same group turned to the hybrid perovskite of MASnI_3_ as channel materials in FET [87]. Apart from those two groups’ works, the perovskite of PbZr_0.52_Ti_0.48_O_3_ (PZT) was used as channel materials in ferroelectric FETs on two occasions [14,88]. Once, the perovskite structured material of KTaO_3_ was used as the semiconductor in FET [89]. After 2011, the same group tried the same perovskite material with a different dielectric layer in some works [90,91]. Table 1 summarizes the properties of FETs reported in these early works.

In the initial perovskite FET studies, tin-based double perovskites were used multiple times. The on/off ratios of the (PEA)_2_SnI_4_ perovskite-based FETs lied around the reasonable value of 10^5^ with the hole mobilities between 0.2 and 3 cm^2^V^−1^s^−1^, but the threshold voltages of those p-type devices were dissimilar and ranged from −1.7 V to −30 V. These studies were not conclusive enough to find the best perovskite materials for the FETs, but these literatures were the foundations for the future explorations in the field.

### 3.4. Recent Works (After 2012) on Perovskite FETs

After the introduction of perovskite materials in solar cells [12,93], a huge number of perovskite materials have been used as the semiconducting active materials in the fabrication of solar cells. In consequence, there is a substantial increase in the number of these since the first MAPbI_3_ channel-based perovskite FET [94]. All the reported perovskite-based thin-film FET studies in the literature after 2012 are classified into three different categories based on the perovskite material types.

#### 3.4.1. Hybrid (Organic Inorganic) Perovskite FETs

The hybrid halide perovskites were used in most of the thin film FET studies. Optoelectronic and material properties of tin- and lead-based hybrid perovskites with different organic cations were analyzed and MAPbI_3_ was found as the most promising hybrid perovskite semiconductor than MASnI_3_, HC(NH_2_)_2_SnI_3_ and HC(NH_2_)_2_PbI_3_ [95]. Having the organic cations makes the device fabrication easier with single hybrid perovskite materials [84]. However, there are concerns over the low carrier mobility and degradation to the moisture and the heat of these perovskites, and the improvements should be made through further research works [96]. The reported hybrid perovskite-based FET literatures after 2012 and their characteristics are summarized in Table 2.

#### 3.4.2. All-Inorganic Perovskite FETs

CsPbBr_3_ is the most dominant of all inorganic perovskite materials, which was studied in [89] many different crystal structures in FET research works. All-inorganic perovskites loaded with CsPbBr_3_ quantum dots and organic rubrene semiconducting sheets were used, and optical and electronic characteristics were analyzed by Youn et al. [132]. In 2019, a phototransistor was fabricated with the semiconducting channels of all inorganic CsPbBr_3_, and the hole mobility was found to be 0.02 cm^2^s^−1^V^−1^ and 0.34 cm^2^s^−1^V^−1^ in dark and illuminated conditions, respectively, with an excellent ambipolarity [133]. The high carrier mobility in the all-inorganic perovskite is improved as the charge carrier mobility is enhanced by the incorporation of inorganic cations of Rb and Cs [134]. Interestingly, an all-inorganic oxide perovskite of KTaO_3_ n-channel was fabricated by Ueno et al. [89]. The thin-film FET studies based on all-inorganic perovskite channels in the literature after 2012 are summarized in the Table 3.

#### 3.4.3. Double and Triple Perovskite FETs

However, the charge carrier mobility is affected by the interlayer distance in these types of perovskites, and the number of FET studies based on double or triple perovskites has increased recently on account of their greater stability than the other two types [146]. Perhaps the earliest study of FET was performed based on the double perovskite of (PEA)_2_SnI_4_ [15]. Then, the various double and triple perovskite materials were used in different forms for thin-film FET applications. Cs_2_AgBiBr_6_ is one of the most popular materials for solar PV applications these days. The Cs_2_AgBiBr_6_ thin-film FETs were studied, and the dominance of the grain boundaries in gating and conduction has been reported. The hole mobility values of Cs_2_AgBiBr_6_ thin-film FETs were shown to strongly depend on the grain boundaries [147,148]. The FET literatures based on the double and triple perovskites and their features after 2012 are summarized in Table 4.

### 3.5. Single Crystalline Perovskite FETs

In a polycrystalline semiconductor material, the presence of grains and grain boundaries cause to the screening effect in the field effect mobility [37]. However, this effect, which is generated by the heat in the polycrystalline material, is denied in the device applications of single crystalline FET [37]. The trap density of single-crystalline perovskites was seen to be lower than polycrystalline perovskites in [164]. The tunnel junction formations occur due to the passive charge accumulations in the grain boundaries [97] in the polycrystalline materials. So, the grain boundary effects are eliminated in single crystalline perovskite FETs, which are less defective than the polycrystalline FETs. However, the poor stability of the single crystalline perovskite materials is a drawback in commercial device applications. The charge transport mechanism in MAPbI_3_ in single crystalline semiconductors was compared with the poly crystalline FET in [11]. The effect of grain boundaries was reduced in the single crystalline transistor, and the electrical characteristics were recorded to be better in single crystalline FET [165]. Electrochemical reactions of Au electrodes in a CH_3_NH_3_PbBr_3_ single-crystal-based FET were investigated in [108]. The single MaPbI_3_ crystal’s photo-generated carrier diffusion was found to be lesser at lower temperatures in [114].

The electrical characteristics of the BGTC device (Figure 7d) and BGBC device (Figure 7e) were compared in [68]. The characteristics of the different single crystalline hybrid halide MAPbI_3_, MAPbBr_3_ and MAPCl_3_ perovskites were compared using the fluorescent microscopic images of Figure 7a–c, respectively. The BGBC (Figure 7g) configuration showed better mobility distribution than the BGTC (Figure 7e) configuration of the MAPbI_3_ single-crystal-channel-based FET. The bottom gate bottom contact device with the single crystalline MAPbI_3_ semiconductor channel-based FET showed the best transfer and output characteristics compared to the other five types of devices [68].

### 3.6. Perovskite FETs with Nanostructured Channel

Perovskite nanomaterials are immensely used in device applications. In FETs, the -perovskite nanomaterial is used in two different types: (1) single nanostructure FETs and (2) thin-film FETs made up of perovskite nanostructures. The single nanostructure FETs consist of a single nanostructure material as a channel. CsPbX_3_ nanowires have received significant interest as a material for optoelectronic applications, including flexible light detectors [166,167]. In the recent past, a considerable number of reports on the all inorganic CsPbX_3_ nanostructure-based FETs have been published, and the characteristics of CsPbX_3_ (X = Cl, Br or I) are depicted in Figure 8a,b.

Meng Y et al. reported a FET with a CsPbBr_3_ nanowire core and MoO_3_ shell nanowire schematic of the nanowire as the channel [37] with a thickness of 1 to 10 nm. The on-current, threshold voltage and the hole mobility of the FET was strongly dependent on the shell thickness. The hole mobility of the nanowire increased from 1.5 cm^2^V^−1^s^−1^ for the pristine CsPbBr_3_ nanowire to 23.3 cm^2^V^−1^s^−1^ for the CsPbBr_3_ nanowire with a 10 nm thick MoO_3_ shell [37].

Nanostructured thin-film FETs consist of a colloidal thin-film channel made up of perovskite nanostructures. The device structure, images of nanostructures used and their transfer curves are given in Figure 9. A bilayer thin-film made up of CsPbBr_3_ quantum dots and dinaphtho[2,3-b:2′,3′-f]thieno[3,2-b]thiophen (DNTT) was tested for FETs (Figure 9a) under dark and light conditions [168]. The transfer characteristic curve of the FET was shown to highly depend on the light intensity, as illustrated in Figure 9b. It was clearly shown that the FET could be turned on and off via shining a light at an intensity as low as 1 mWcm^−2^. The same group reported a bilayer FET (Figure 9c) based on CsPbI_3_ nanorod and 2,7-Dioctyl benzothieno[3,2-b] benzothiophene (C8BTBT) [169]. The bilayer nanorod FETs were tested for photo-detectors, and the threshold voltage of the FET was increased with increasing light intensity from 0.05 mWcm^−2^ to 10 mWcm^−2^, as shown in Figure 9e. A FET was reported made-up of CsPbBr_3_ perovskite NCs terminated with short ligands (Figure 9f–h) in [83]. The FETs were tested under different temperatures, and a positive shift was also observed in the transfer curves with increasing temperature (Figure 9i,j). The hole mobility derived from both forward and backward scanning during the cooling and heating cycle of these FETs exhibited a temperature hysteresis effect.

## 4. Applications of Perovskite FETs

The extremely scalable and quite inexpensive synthetic and fabrication techniques of these materials under ambient conditions are the major advantages [52]. Perovskite-based solar cells are on their way to be commercialized, and perovskite FET applications are not far from being commercialized. Thin-film FETs have been extensively studied based on the perovskite semiconducting channels and are used in numerous different applications [170]. The optoelectronic properties of the perovskite materials [171] are mainly caused by the studies of these applications, which are classified into five major categories. The perovskite materials are being used as semiconducting channels to study the thin-film FETs [73] to perform the following categorized device applications.

### 4.1. FET Platforms as Tool for Evaluating the Carrier Mobility

FET-based platforms are widely used to study the electron and hole mobilities of a material. As discussed in Section 3.2.3, the mobility can be calculated using the sub-threshold swing of the transfer curve of a FET. Since it is a simple technique, many researchers use this method to evaluate the mobility of the perovskite channel with different material composition and at different operating conditions. Qingfeng D et al. reported hole mobility values of 164 (±25) cm^2^V^−1^s^−1^ using a space charge limit current and 105 (±35) cm^2^V^−1^s^−1^ using hall effect measurements in a single crystalline solution-grown MAPbI_3_ single crystal [172]. In another work, extremely high mobilities of 500–800 cm^2^V^−1^s^−1^ were reported using optical pumped THz spectroscopy on a single crystal MAPbI_3_ perovskite. However, very low mobilities of 2–20 cm^2^V^−1^s^−1^ were reported for MAPbI_3_ single-crystal perovskites using FETs [68]. This variation could be due to several assumptions made while fixing the characteristic curves. Particularly, in FETs, the channel is assumed to be a uniform film with a fixed length and width. It also neglects any losses at the grain boundaries and channel–metal electrode interfaces. Despite these limitations, FET platforms are being widely used as a reliable method to compare the mobilities at different material compositions [68], temperatures [83,173] and testing conditions.

### 4.2. Photo FETs

Photo FETs are the FET devices used for detecting and quantifying light. Due to the excellent photosensitivity of perovskite FETs, they are widely used as photo FETs. One type of carrier is immobilized due to the incident light, and the other type of carrier contributes to the photocurrent in perovskite photo FETs [27]. When the channel is exposed to the light, the career density changes rapidly, and FET also shows a significant change in the transfer curves. In photo FETs, the light can be identified by comparing the on–off ratio, on-current, shift in threshold voltage or change in hysteresis [153] in both light and dark conditions. The dependence of the electrical characteristics of the device and the on/off functions with regards to the illumination and dark conditions were illustrated in [153].

One of the largest values of photo responsivity was achieved as 320 AW^−1^ with a MAPbI_3_ channel-based phototransistor, and the output characteristics under the dark and illumination were illustrated completely in [97]. The photoconductive gain G of a perovskite photo FET is given by the following equation [97]:$$G=\frac{\tau_{\mathrm{life}}}{L^{2}/\left(\mu .V_{\mathrm{ds}}\right)}$$
where $\tau_{\mathrm{life}}$ is the recombination time of the carriers and L is the perovskite channel length, μ is the carrier mobility and V_ds_ is the applied source drain voltage. Generally, the denominator $L^{2}/\left(\mu .V_{\mathrm{ds}}\right)$ is the transit time $\tau_{\mathrm{tran}}$ of the charge carriers. In 2015, F. Li, C. Ma, H. Wang et al. reported a MAPbI_3_ channel based ambiolar photo FET (Figure 10a) with estimated photoconductive gain in the range of 10 to 10^2^. The photo FET showed an excellent ambipolarity with p-type and n-type transfer characteristics. The transfer characteristics of the device under the dark and illuminated conditions are given in the Figure 10b [97].

The photo-modulated hysteresis behavior was observed in a MAPbI_3_-based photo FET (Figure 10c). The anticlockwise hysteresis effect of the photo FET was narrowed by the monochromatic light, as shown in the Figure 10d [99]. A flexible photo FET was reported using the same MAPbI_3_ perovskite channel in [117]. It was also shown that the MAPbI_3_ nanowires in the channels improved the transfer characteristics of the pure polymer-based phototransistors [111]. Despite the superior photoconductive gain and possibility of a photo-modulated current in the perovskite channel, the application of perovskite photo FET has not been explored much in the past. Table 5 summarizes the perovskite semiconductor-material-based photo FETs reported so far.

### 4.3. Photo Detectors

Perovskite materials are widely used as the sensing material in many two-terminal photodetectors. However, the three-terminal devices perform better than two-terminal devices as they reduce the noise during the built-in amplification process of the electrical signals induced by the photo detection [176]. The combination of the optical and electronic properties in perovskite materials determines the performance of the thin-film FET-based photo detectors [177]. In signal-processing technology, the fundamental function of the photo detectors is the very important conversion of the optical signal to an electrical current. FASnI_3_ showed a high responsivity in the range of 10^5^ AW^−1^ in [103].

The performance of the photo detector device was analyzed by the following calculations in [105,178], the responsivity R of the device was given in terms of irradiance of E_e_:$$R=\frac{I_{P}}{{\mathrm{WLE}}_{e}}$$
where the Ip is the difference between the light photo current and dark photo current, W is the channel width and L is the channel length. The detectivity D^∗^ is given by the responsivity R, channel area A, the charge of electron and the dark current I_DS_ as in the following equation [174]:$$D^{*}=\frac{{\mathrm{RA}}^{1/2}}{(2{\mathrm{eI}}_{\mathrm{DS}}{)}^{1/2}}$$

Generally, perovskite materials are used as the absorber materials with oxide semiconductors [179], carbon nanotubes [151] and graphene [180] channel-based FETs. Table 6 summarizes the works reported on perovskite channel-based FET photodetectors, along with their photoresponsivity and other figure of merit values.

### 4.4. LEFETS

The exceptional electroluminescence of perovskites enables the application of light-emitting FETs (LEFETs) based on perovskite semiconducting channels (Table 7). In 2015, CH_3_NH_3_PbI_3_ was used as the channel material to fabricate a LEFET [20]. The BGTC FET device with uniform CH_3_NH_3_PbI_3_ layer as the channel on a Si/SiO_2_ substrate and Au electrodes of source and drain was reported (As in Figure 11a) and tested at low temperatures. The transfer and output characteristics under positive and negative gate bias were tested and showed an ambipolar charge transport as depicted in Figure 11b,c respectively. The light emitting operation functioned with the gate-assisted electroluminescence at low temperatures below 200 K. The symmetric ambipolar nature of the device allows the electrons and holes which are injected from the opposite electrodes to form a narrow radiative emission zone in the middle of the dielectric and channel interface.

Later in 2018, the same group reported that the light emission in the FET was observed during the ambipolar regime in the temperature range of 78–178 K, as there was a reduction in the ion-induced gate screening effect [104]. The same group followed with solutions to the ionic drift and cation polarization in the CH_3_NH_3_PbI_3_ thin-film FETs [102]. They tested the same FET devices under different operational configurations as in Figure 11d–f. However, in the configurations, the DC-driven current induced methyl ammonium cation polarization and a strong ion migration in the perovskite channel. As a result, when a high-frequency AC-driven gate field was applied, the LEFET showed better performances. However, the reduction in temperature minimized the effect of ion polarization, and the hysteresis effect was reduced [102]. Although the light emission occurred in proximity to the source and drain electrodes in the perovskite channel, as the bright electroluminescence was observed there, variations were observed in the spatial positions of light emission due to the differences in the applied drain voltage and gate voltage with regards to the gate voltage, as shown in Figure 11d–f. In addition to that, the LEFET functioned simultaneously as a switch and as a light-emitting device [102]. MAPbI_3_ is the most prominent material in light-emitting FETs. However, the on/off ratios of these FETs suggest that improvements should be made as the performances with emission peak wavelength at 800 nm was achieved, as shown Figure 11g,h.

D’Innocenzo et al. demonstrated that the light-emitting properties of hybrid lead–halide perovskites could be fine-tuned [186]. It was concluded that the larger the crystallites, the smaller the band gap becomes and the more the lifetime of the light emission increases. The stripe length method of optical gain measurement was used to calculate the output intensity I by making the spatial profile of the photo excitation into stripes:$$I\left(\lambda ,\;l\right)=\frac{A\left(\lambda \right)I_{P}}{g\left(\lambda \right)}e^{g\left(\lambda \right)l}-1$$
where A is a constant for spontaneous emission cross-section, $I_{P}$ is the intensity of the light, g is the gain coefficient and λ is the wavelength of the light.

**Table 7 nanomaterials-12-02396-t007:** The reported light-emitting FET devices based on the perovskite structured channels inthe literature.

Perovskite	Hole Mobility (μ) cm^2^s^−1^V^−1^	On/Off Ratio	Wavelength Peak nm	Ref
MAPbI_3_	0.072 at 78 K 0.00002 at 300 K	10^6^	800	[20]
MAPbI_3_	0.025 at 78 K	1000	783	[102]
MAPbI_3_	0.025 at 78 K	1000	783	[187]
NFPI_7_	20 at 300 K	10^6^	800	[188]
MAPbI_3_ Microplate	4 at 77 K	10^3^	800	[173]
CsPbBr3 NC/Poly fluorine	3.3 at 250 K 5 at 100 K	10^5^	500	[136]

NFPI_7_: Perovskite deposited using 1-naphthylmethylamine iodide (NMAI), formamidinium iodide (FAI), and PbI_2_ with a molar ratio of 2:1:2 dissolved in N,N-dimethylformamide (DMF).

### 4.5. Ferroelectric RAM

Oxide perovskites showed significant progress in the studies of FETs as Ferroelectric RAM. The first ever perovskite-based FET study itself was a ferroelectric FET [14]. By reading the conductance of the semiconducting channel in a perovskite FET, a non-destructive memory read-out is permitted [14]. The electrically switchable ferroelectric spontaneous polarization enable the ferroelectric random access memory [189]. In a ferroelectric FET, the current between the source and drain electrodes is controlled by the gate voltage and the ferroelectric conductance of the channel layer, and when this conductance is higher, the drain current flows along the interface of the ferroelectric and insulator layers [75]. Unlike the other perovskite FET applications, the hysteresis effect of the perovskite FET is a positive one in applications of ferroelectric RAM for obtaining non-volatile random access memory [31].

The ferroelectric hysteresis loop in the first ever perovskite-FET-based RAM study [14] was relatively square; hence, the remnant polarization is quite high for all perovskite ferroelectric FET memory devices with a TGTC structure [14]. The corresponding band diagrams of the ferroelectric FET were discussed by S. Mathews et al. [31]. The polarization directions with the up and down movements of the hole carriers were noticed in the ferroelectric layer, and the suppressed and the accumulated states of the hole carriers in the channel layer were denoted in the study [31]. A similar trend in the transfer characteristic curve of the PZT channel-based ferroelectric FET [64] was observed by Lee B.Y. et al. The clockwise hysteresis loop indicated that the source drain current I_ds_ is controlled by the ferroelectricity PZT channel layer. A completely new perovskite, GdNi_0.2_Fe_0.8_O_3_ (GFNO), was used as a channel material in a perovskite-FET-based ferroelectric RAM in 2020 [190]. A MoS_2_ monolayer was used to tune the electrical characteristics of the ferroelectric perovskite FET, and the non-volatile device impressively retained the transistor state even in the absence of electrical bias [190].

A review of flexible ferroelectric perovskite thin-film studies was conducted focusing on oxide perovskite materials in 2020 by Gao et al. [31].

## 5. Challenges

Despite their attractive applications, perovskite FETs have a lot of challenges to overcome in order to become a dominant player in the industry. The environmental and operational stability issues of perovskite materials are well-studied and addressed in solar PV research works [33]. The current hysteresis is known to be one of the main challenges in perovskite-based solar cells and FETs. The hysteresis effect was also reported in both solar PV devices [90] and FETs. However, hysteresis in FETs is approached in a different way. For memory applications, the hysteresis effect is essential [15,67,141] while the hysteresis effect is a hindrance in all other applications [16,128]. We first discuss the hysteresis effect and approaches to reduce hysteresis for FETs in detail

### 5.1. Hysteresis in the Transfer Curves of the PFETs

The main reason for hysteresis in both transfer and output I-V measurements is known to be the accumulation of ions near the gate region or at the grain boundaries [191]. The applied gate voltage alters the distribution of ions, particularly the halide anions in the perovskite channel, which also redistribute the defects in the films or crystals. Generally, two types of approaches are used for suppressing hysteresis: (1) device structure engineering and (2) channel material engineering. A few recent works have reported a reduction in the hysteresis effect of perovskite FETs by engineering the gate–channel and channel–electrode interfaces. Matsushima et al. reported a work on reducing the hysteresis of a FET with (a PEA)_2_SnI_4_ perovskite channel. The step-by-step reduction in hysteresis with the addition of a self-assembled monolayer containing ammonium iodide terminal groups in between the gate and channel and a MoO_x_ thin-film between the channel and source-drain electrodes was reported, as shown in Figure 12a–c. The threshold voltage decreased from −34 (±4) V to −22 (±2) V and the hole mobility increased from 0.53 (±0.24) to 12 (±1) cm^2^V^−1^s^−1^ in the devices with an interface layer at both the gate–channel and channel–electrode interfaces. Doping the channel material is another approach to reduce the hysteresis effect [90].

Few such works report reducing the hysteresis in FETs using amine-based additives to the perovskite channel materials, achieving a cross-link between the grain boundaries. The most recent work on the phototransistor was performed based on the (PEA)_2_SnI_4_ double perovskite [175]. They use the Lewis base adduct method to enhance the performance of the perovskite-based phototransistor. Non-volatile urea was applied to the perovskite grain boundaries on the inside of the lattice as a bi-functional additive. The urea exerts a passivation effect on the grain boundaries, the region rich in charge carrier traps. Hence, the screening effect was reduced between the grains and the hysteresis effect was reduced [175] as the reduced gap between the forward and backward sweeps of the transfer curve of the 2% urea-added perovskite channel-based FET (Figure 12e), with a better stability, as shown in Figure 12f, than the pristine channel-based FET’s gap (Figure 12e). As a result, a small dual-sweep hysteresis was achieved with an improved mobility of 4.2 cm^2^V^−1^s^−1^ and an on/off ratio of 6.7 × 10^5^ [175].

Not long ago, the hysteresis effect in a p-type MASnX_3_ perovskite FET was exceptionally reduced by the halide anion engineering [131] of the hybrid halide perovskite channel using bromide and chloride co-substitutions or partial iodide. In this study, it was observed that the co-substitution of the anions enhanced the quality of the film and reduced the vacancy defects as well. As a result, the hysteresis effect was significantly reduced to a negligible level (0.1 V) in the transfer characteristics curves of Cl^−^ and Br^−^ incorporated MAPBI_3_ channels, whereas the pristine MASnI_3_ channel showed a notable 1.5 V difference between the forward and backward sweeps in the transfer curves. The mobility variation ratio of the forward and backward sweeps was reduced to 20% for Cl- and Br-incorporated MAPBI_3_-based FETs from the 45% of the pristine MASnI_3_ channel FET [131]. Y. Liu et al. reported that the hysteresis effect increases notably with the addition of SnI_4_ to the (PEA)_2_SnI_4_ channel; nevertheless, an enhancement was observed in the device current and on/off ratio [162]. In addition, it was assumed that the presence of extra I^-^ ions in the channel caused the larger hysteresis effect [162].

### 5.2. Operational and Chemical Stability

In the recent past, the use of all-inorganic, double and triple perovskite materials for PV applications received significant attention due to their operational and chemical stability [50,167,192,193,194]. Perovskite FETs remain in the research stage, as the commercialization of these devices faces the uphill task of overcoming their poor stability due to environmental exposure [195]. Enhancement was observed in the structural and the operational stability of the hybrid halide–perovskite-based optoelectronic devices analyzed in [189]. A MAPbI_3_ channel-based FET was fabricated to test the performance of the device over time. The on-drain source current of the device was found to be gradually decreasing with time in days, even in the dark conditions with a relative humidity between 40% and 60% (As depicted in Figure 13b) [196]. Encapsulation of CYTOP was found as a solution to avoid degradation with the time in (PEA)_2_SnI_4_-based FETs [196]. The initial mobility of 0.1 cm^2^V^−1^s^−1^ of (PEA)_2_SnI_4_-based FETs faded within a day to 0, whereas the (4-Tm)_2_SnI_4_-based FET’s initial mobility of 1 cm^2^V^−1^s^−1^ reduced to 0.1 cm^2^V^−1^s^−1^ after 30 days because the larger organic cations provide more stability in the environment [158], as shown in Figure 13c. According to Figure 13a, when the ambient temperature increases, the drain source current of the MAPbI_3_-based FET decreases drastically [97].

### 5.3. Potential of Perovskite FETs and Future Scope

As the perovskite semiconducting channels show exceptional ambipolarity, the complementary inverters are beginning to attract the attention of researchers in the advancement of practical applications of perovskite FETs. A triple-cation Cs_x_(MA_0.1_7FA_0.83_)_1−x_Pb(Br_0.17_I_0.83_)_3_ perovskite played an exemplary ambipolar nature with the variation in the stoichiometric content of Cs cations. A conventional type of inverter was constructed with two perovskite FETs (n- and p-types). On account of the tuneable channel conductivity for electrons and holes, unlike unipolar inverters, perovskite FET-based inverters can be operated in the first and third quadrants with maximum gains of 21 and 23, respectively [76]. H. Zhu et al. found that MASnIxBryCl_1−x−y_-based FETs operated in an ideal enhancement mode with a threshold voltage of 0V. Hence, an inverter was studied by using a p-type FASnX_3_-based FET along with n-type Indium Gallium Zinc oxide (IGZO) [131]. These studies lead to a pathway for future developments in inverters based on perovskite FETs. B. Jeong reported a TGBC-structured CsPbBr_3_ perovskite FET along with the ferroelectric layer of P(VDF-TrFE) [197]. The electrical conductivity of the perovskite’s active channel was utilized by the ferroelectric gate to enable multi-state conductance. It was concluded that a perovskite FET with a ferroelectric gate could be a potential gateway to various ionic–electronic mixed conductors, potentially realizing hardware for future artificial intelligence [197].

All the reported perovskite channel-based ferroelectric FETs are reported in the Table 8.

## 6. Conclusions

As we summarised and discussed in this work, perovskite semiconductor materials are considered as unique materials due to their chemical and electrical properties. Their use as the channel material in FETs received increasing attention for their unique electro-optical properties and wide range of applications. Based on the literature, MAPbX_3_ (X = Cl, Br, I) and CsPbBr_3_ are the most-studied hybrid and all-inorganic perovskites for FET applications, respectively. Apart from these materials, the hybrid double-perovskite of (PEA)_2_SnI_4_ was also studied for FET application. The FET platform is widely used for evaluating the mobility of the material. Figure 14 summarizes all the mobility values reported for the FETs with selected perovskite materials throughout the years.

MAPbX_3_ (X = Cl, Br, I), and CsPbBr_3_ are the popular materials used for photo FET applications due to their improved PV properties. Photodetectors are generally a FET with two layered channels where graphene was the dominant channel at the bottom layer to facilitate a photosensitive perovskite channel on top. Numerous perovskite materials including BA_2_PbI_4_, MAPbBr_3_, FASnI_3_, (PEA)_2_SnI_4_ were used as the photosensitive layer. For LEFET applications, the most dominant material used in the channel is MAPbI_3_. The emission peak of these reported LEFET was around 800 nm which is in the near infra-red region. However, the tuneable emission peak with gate voltage is the promising sign in these devices. Also, electroluminescence was generally observed at low temperatures. Finally, we go into the FETs with perovskite channels and ferroelectric layers, which are used in the studies of random access memory (RAM).

Despite numerous advantages of the perovskite materials, and diverse unique device applications, still the perovskite FETs have received relatively poor attention. The flexible ionic structure of the perovskite materials makes it more complex for an environmentally and operationally stable FET device. The hysteresis effect is one of the major drawbacks in these devices and a few recent works proposed two different mechanisms to overcome this issue: (1) using an isolated gate-channel and electrode-channel interfaces to reduce the ion migration and (2) using dopants to immobilize the ions in the channel material. In the case of ferroelectric RAM, we have to mention that the hysteresis effect is used as a positive effect as well. However, it should be noted that in FETs, the channel material experiences a high gate electric field and hence, proper mechanisms to be developed to improve the operational stability of perovskite materials to fabricate the channel of a FET.

The extensive range of perovskite materials can also be tested for stable FET devices for various applications. Again, the operational stability of these materials has to be studied at the channel of the FET. Moreover, the gating mechanism of perovskite FETs is also not fully explored. The role of the bulk channel, the channel–electrode junctions and the role of grain size and grain boundaries in electrostatic and optic gating need to be studied in detail to improve the figure of merits of perovskite FET devices. Gate-controlled electroluminescence spectra in perovskite FETs is a promising gateway to continuous controllable wavelength lasers or light-emitting devices. Their applications in high-frequency devices make them a potential candidate for communication devices and built-in optical amplifiers. More focused research on hysteresis issues, operational and environmental stability and gating mechanisms under light and dark conditions will lead these unique devices into the next stage and inch them towards commercial applications.

## Figures and Tables

**Figure 1 nanomaterials-12-02396-f001:**
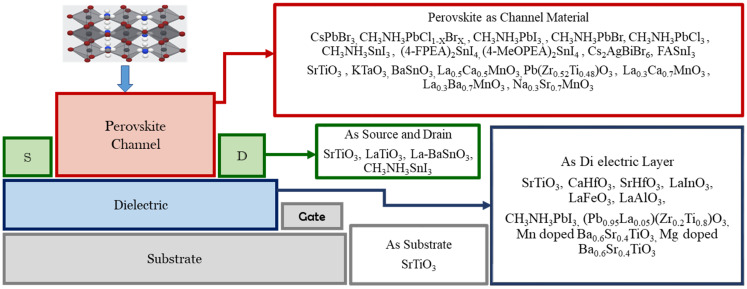
An enormous number of perovskite materials are being used to perform different functions in the thin-film FET literature.

**Figure 2 nanomaterials-12-02396-f002:**
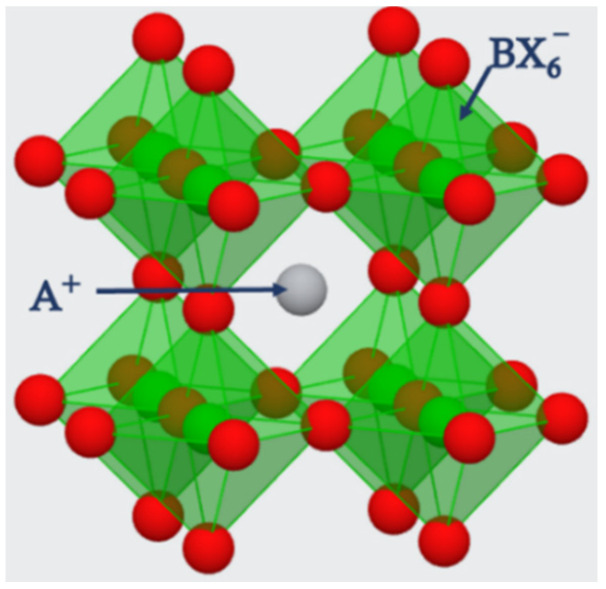
The corner sharing 3D crystal structure of ABX_3_ perovskite. A⁺ ions embedding the voids of octahedral ${\mathrm{BX}}_{6}^{-}$ by sitting in their cavities.

**Figure 3 nanomaterials-12-02396-f003:**
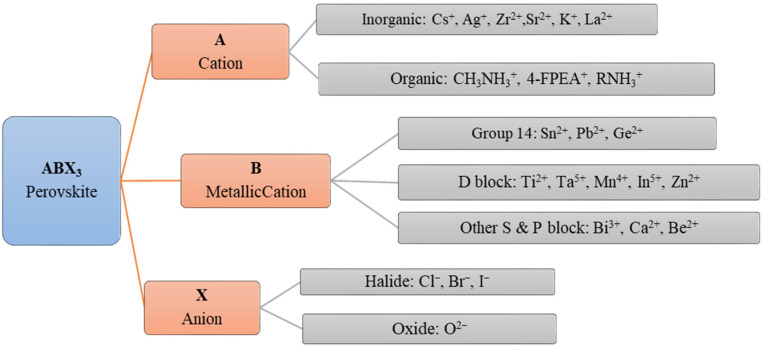
Classification of the different ionic components formulating the compositions of ABX_3_ 3D perovskites for semiconducting channel materials.

**Figure 4 nanomaterials-12-02396-f004:**
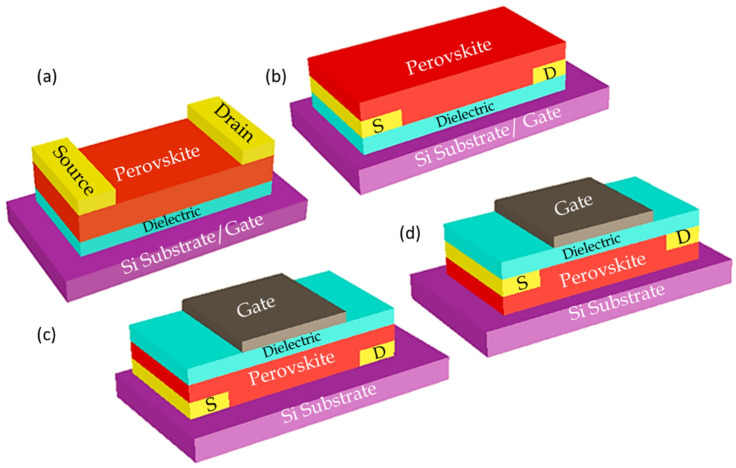
Device structures of perovskite FETs with four different globally recognized configurations (**a**) bottom gate top contact (BGTC), (**b**) bottom gate bottom contact (BGBC), (**c**) top gate bottom contact (TGBC) and (**d**) top gate top contact (TGTC).

**Figure 5 nanomaterials-12-02396-f005:**
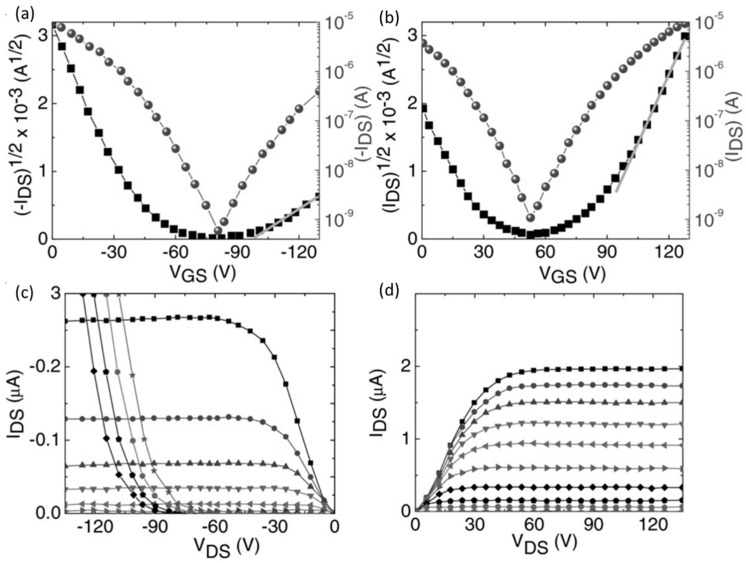
The transfer characteristic curves of the (**a**) p-channel and (**b**) n-channel hybrid perovskite-channel-based ambipolar FET. The output characteristics (**c**) p-channel and (**d**) n-channel of a triple cation perovskite-channel-based ambipolar FET. Reproduced with the permission from [76]. Copyright 2016 WILEY-VCH Verlag GmbH & Co. KGaA, Weinheim.

**Figure 6 nanomaterials-12-02396-f006:**
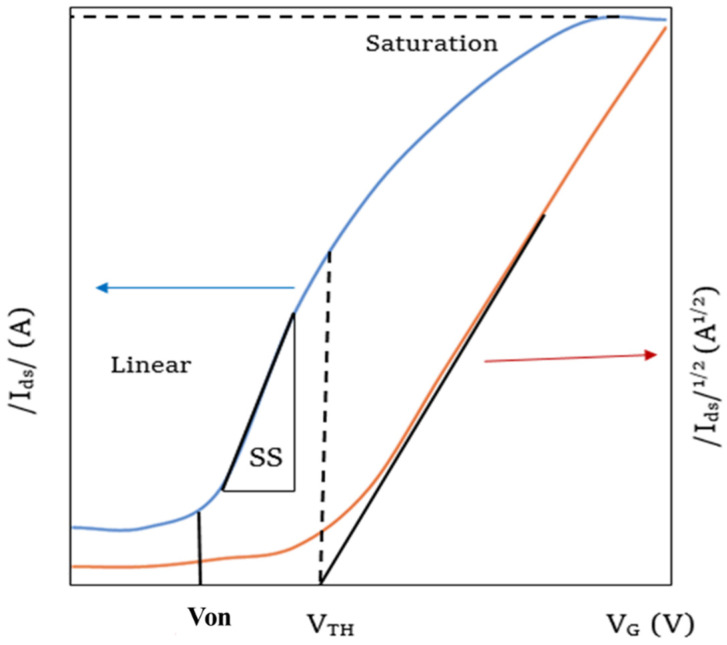
Transfer characteristics curve of a perovskite thin film FET with linear and saturated regimes.

**Figure 7 nanomaterials-12-02396-f007:**
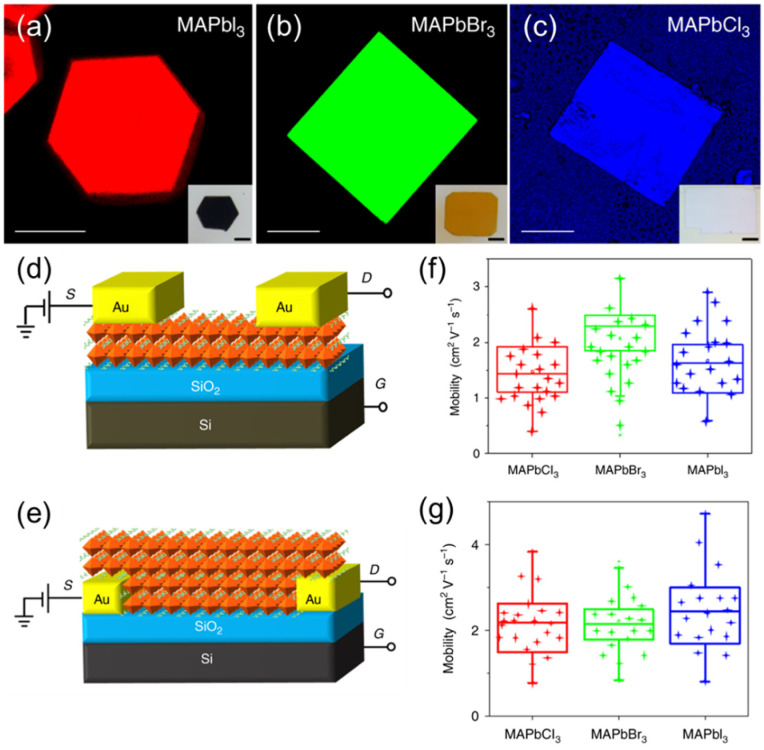
Fluorescence microscopy images of (**a**) MAPbI_3_, (**b**) MAPbBr_3_ and (**c**) MAPbCl_3_ single crystals (excited with a pulsed lasers with wavelengths of 450, 473 and 405 nm, respectively) with a scale bar of 100 μm. Inset: optical images of the corresponding single crystals with a scale bar of 200 μm. Schematic of the (**d**) bottom gate top contact (BGTC) and (**e**) bottom gate bottom contact (BGBC) device with the tested perovskite single crystal as semiconductor layer and the hole mobility distribution of (20 devices) each halide perovskite under (**f**) BGTC and (**g**) BGBC device structure [68], Copyright 2018 Springer Nature.

**Figure 8 nanomaterials-12-02396-f008:**
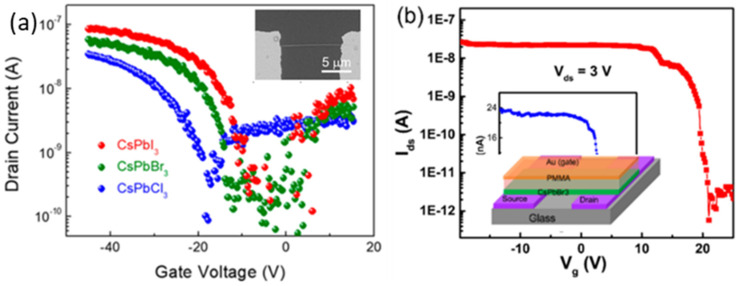
(**a**) Transfer characteristics of the typical single VLS−grown CsPbX_3_ (X = Cl, Br or I) NW FETs using logarithm y-coordinate. Inset shows the SEM image of the as-fabricated NW PD (**b**) Semilog CsPbBr_3_ plot of the gate transfer characteristic Ids-Vg measured at Vds = 3.0 V and swipe rate 1.1 V/s. Inset shows the same plot in linear scale. Reproduced with permission from [166]. Copyright 2020, American Chemical Society.

**Figure 9 nanomaterials-12-02396-f009:**
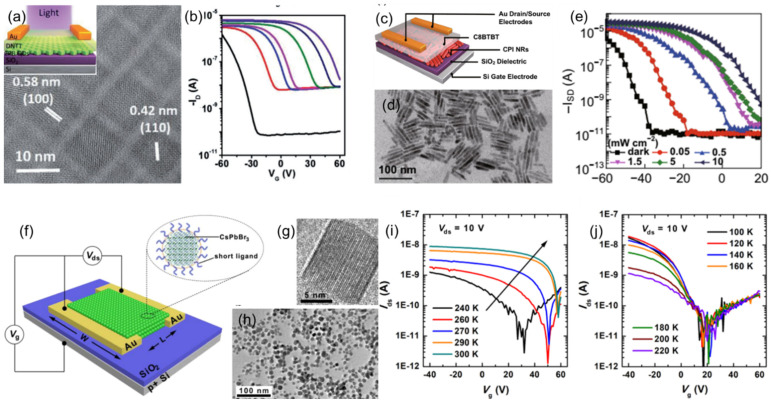
(**a**) HRTEM image of CsPbBr_3_ quantum dots (inset: device schematic of the FET) and (**b**) transfer characteristic curves under different illumination conditions. Reproduced with the permission from (**c**) [168]. Copyright 2017, Wiley-VCH Verlag GmbH & Co. KGaA, Weinheim. (**c**) Device schematic of the bilayer FET made-up of CsPbI3 nanorods and C8BTBT, (**d**) TEM image of CsPbI_3_ nanorods (**e**) transfer curves of the FETs under different light intensities [169]. Copyright 2018, Springer Nature. (**f**) Device schematic along with FET characteristic connections of the CsPbBr_3_ NCs terminated with short ligands, (**g**) high- and (**h**) low-resolution HRTEM image of the CsPbBr_3_ NCs and transfer characteristic curves of the CsPbBr_3_ NC FETs under different testing temperatures. Reproduced with permission from, transfer characteristics of undoped CsPbBr_3_ NCs FETs (**i**) from 240 to 300 K and (**j**) 100 to 200 K [83]. Copyright 2020, American Chemical Society.

**Figure 10 nanomaterials-12-02396-f010:**
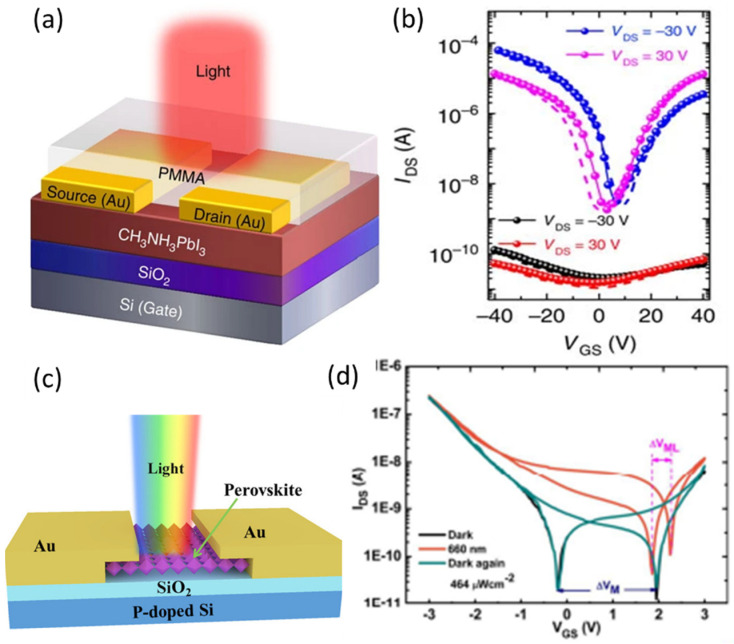
(**a**) Device schematic of CH_3_NH_3_PbI_3_ channel-based phototransistor. (**b**) Transfer characteristics of phototransistor in dark (red and black) and in illumination (blue and magenta) [97]. Copyright 2015, Springer Nature. (**c**) Device structure with MAPbI_3_-based photo FET. (**d**) The hysteresis behaviour and ambipolar transfer characteristics of the photo FET in dark (blue), under 660 nm illumination (orange), under dark again (black). Reproduced with permission from [99]. Copyright 2018, American Chemical Society.

**Figure 11 nanomaterials-12-02396-f011:**
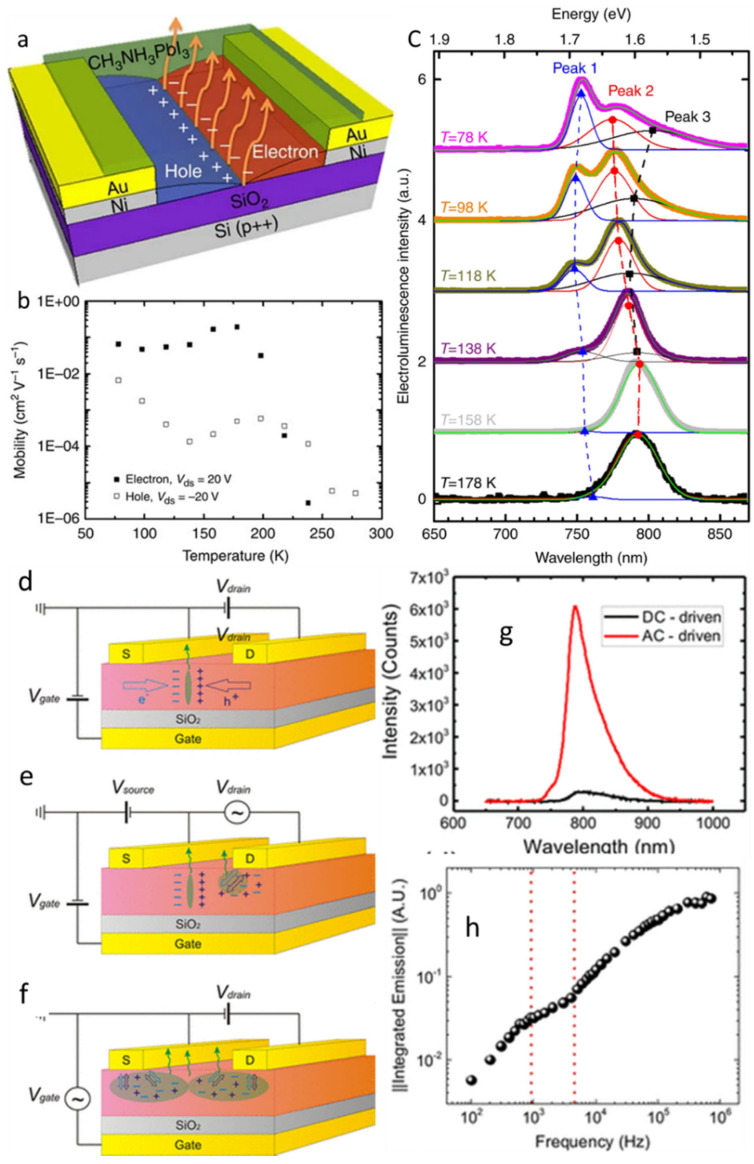
(**a**) The schematic of the bottom gate bottom contact light emitting FET with CH_3_NH_3_PbI_3_ thin-film channel. (**b**) Variation in hole and electron mobilities of the same FETs under different operating temperatures and (**c**) electroluminescence spectra of LEFET under different operating temperatures [20]. Copyright 2015, Springer Nature. The schematics of (**d**) DC-driven gate and drain fields form a thin recombining region. (**e**) AC-driven drain field continuously injects the holes and electrons from the drain; (**f**) both electrons and holes are injected from the source and drain by the applied AC-driven gate voltage used for testing LEFETs. Reproduced with permission from [102]. Copyright 2018 Chemical American Society. (**g**) The variation in electroluminescence under fixed and pulsed gate bias and integrated emission with the frequency of the voltage pulse applied at gate. (**h**) Normalized integrated emission of the electroluminescence peaks versus driving voltage frequency. Reproduced with permission from [179]. Copyright 2017 WILEY-VCH Verlag GmbH & Co. KGaA, Weinheim.

**Figure 12 nanomaterials-12-02396-f012:**
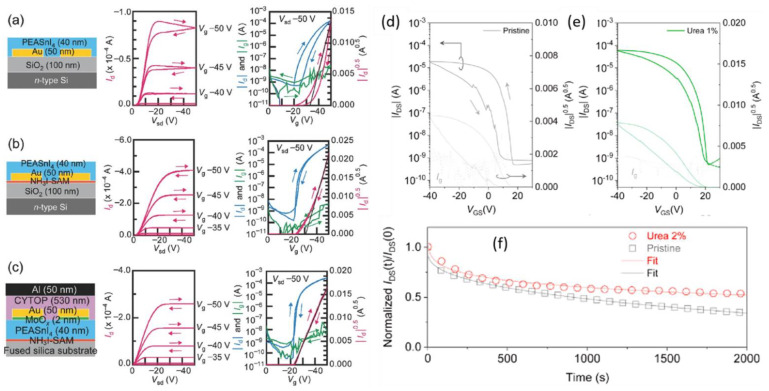
(**a**) Representative output and transfer characteristics of BC/BG transistors (**a**) without Self Assembled Monolayer (SAM), (**b**) with NH_3_I-SAM and (**c**) TC/TG transistors with NH3I-SAM when measured with forward and reverse scans at a scan rate of 5 Vs^−1^. Gate currents (Ig), which are correlated with leakage currents occurring through gate dielectrics, are also shown in each Figure. Ig is not so large compared with Id for all of the transistors. Reproduced with permission from [90]. Copyright 2016 WILEY-VCH Verlag GmbH & Co. KGaA, Weinheim. Transfer characteristics of (**d**) pristine and (**e**) 2% urea-added (PEA)_2_SnI_4_ perovskite-based FETs and (**f**) their current stability over time. Reproduced with permission from [175]. Copyright 2021 Elsevier Ltd.

**Figure 13 nanomaterials-12-02396-f013:**
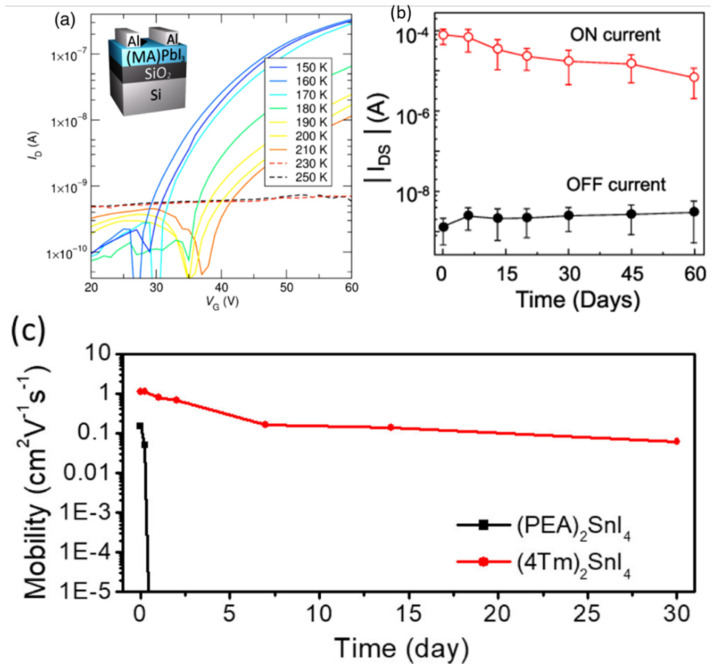
(**a**) Transfer curves of MAPbI_3_-based FET at various temperatures between 150 K and 250 K [94]. (**b**) Evolution of ON and OFF currents as a function of time, where the devices were kept in the dark with a relative humidity between 40% and 60% over a period of 2 months in ambient atmosphere. Reproduced with permission from [196]. Copyright 2020, American Chemical Society. (**c**) Evolution of hole mobilities of the BGTC FET devices based on (PEA)_2_SnI_4_ and (4Tm)_2_SnI_4_ over a long storage time. Reproduced with the permission from [157]. Copyright 2019, American Chemical Society.

**Figure 14 nanomaterials-12-02396-f014:**
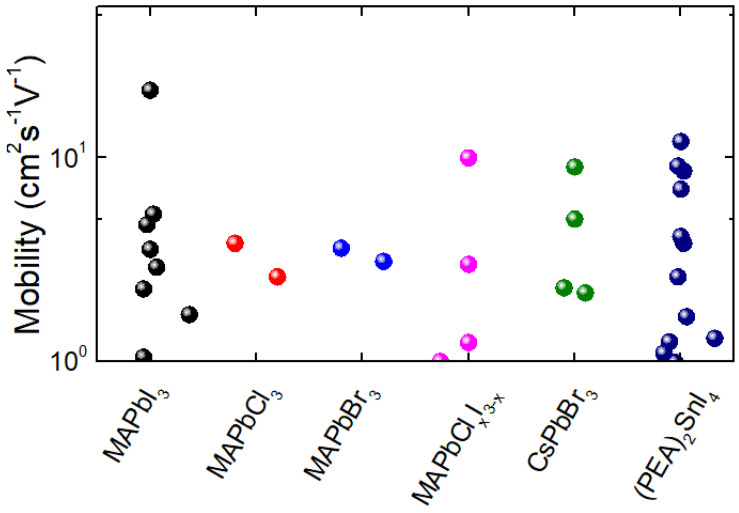
Distribution of hole mobility of various perovskite materials studied using FET structure.

**Table 1 nanomaterials-12-02396-t001:** Early works of thin-film FETs with the perovskite-material-based channels in the literature.

Perovskite Semiconductor	Substrate	Source/Drain	Insulator	Device Structures	μ_e_/μ_h_ [cm^2^ V^−1^s^−1^]	Ion/Off	V_TH_ [V]	Ref
(PEA)_2_SnI_4_	Si	Pd	SiO_2_	BGBC	0.62	10^4^	−30	[15]
(2-PEA)_2_SnI_4_ (3-PEA)_2_SnI_4_ (4-PEA)_2_SnI_4_	Si	Au	SiO_2_	BGBC	0.24 0.51 0.48			[16]
(PEA)_2_SnI_4_	Si	Pd	SiO_2_	BGBC	0.1–0.5	10^4^−10^5^		[92]
(3-MeOPEA)_2_SnI_4_ (4-MeOPEA)_2_SnI_4_	Si		SiO_2_/Polyamide	BGBC	2.6 1.3	10^6^ 10^4^		[84]
(PEA)_2_SnI_4_	Si		SiO_2_	BGBC	0.28	10^5^	−3.2	[83]
KTaO_3_		Au	Al_2_O_3_	BGTC	0.4	10^5^		[89]
(PEA)_2_SnI_4_	Si	Au	SiO_2_	BGTC	0.78	4.2 × 10^5^	−1.7	[85]
MASnI_3_	Si	Au/Pt	SiO_2_	BGTC	0.004	10^3^	−7.2	[87]
PZT	Si	Pt	HfO_2_	TGBC		10^6^		[70]
PZT	LaAlO_3_	La_2−X_Sr_X_CuO_4_	LCMO	TGTC				[14]

Cytop: Amorphous fluorinated polymer; LCMO: La_1−x_Ca_x_MnO_3_.

**Table 2 nanomaterials-12-02396-t002:** The single hybrid organic–inorganic perovskite structured material channel based thin-film FET works in the literature after 2012.

Perovskite Channel	Substrate	Source Drain	Insulator	Device Structures	μ_h_/μ_e_ [cm^2^ V^−1^s^−1^]	Ion/Off	V_TH_	Ref
MAPbI_3_ MAPbI_3−X_C_X_	Si	Au	SiO_2_	BGTC	0.18 (0.17) 1.24 (1.01)	10^4^	20, −5	[97]
(MAPbI_3_)_X_(MAPbBr_3_)_1−X_	Glass	Au/Cr	Cytop	TGBC	21.36 (49.32)			[98]
MAPbI_3_	Si	Au	HfO_2_	BGBC	1.05	10^4^	−1	[99]
MAPbI_3_	Si	ZnO	SiO_2_	BGBC				[100]
MAPbI_3_	Si	Au, Ti	SiO_2_	BGBC				[101]
MAPbI_3_	Si	Au	SiO_2_	BGTC	0.025 (0.11)			[102]
MAPbI_3_	Si	Au Au/MoO_X_	SiO_2_	BGTC	3.55 7.47	10^5^		[103]
MAPbI_3_	ITO	Al	HfO_2_	BGBC	0.02	10^4^		[104]
FASnI_3_	Si	Cr/Au	SiO_2_	BGBC				[105]
MASnI_3_	Si		SiO_2_	BGBC	0.007			[106]
MAPbI_3_/PMMA	Si		Glass	BGTC	0.07			[78]
MAPbI_3_ Nanowire	Si	Ag	SiO_2_/PDMS	BGTC		10^2^	−5	[107]
MAPbI_3_	Si	Cr/Au	SiO_2_	BGBC	15			[108]
MAPbI_3_ MAPbI_3_/ZnO NC	Si	Au/Ti	SiO_2_	BGBC	0.00001 0.00002		−48 −59	[109]
MAPbI_3_	Glass	Al	V_2_O_5_/LiF	TGBC		10^5^		[110]
MAPbI_3_ PDVT 10 MAPbI_3_ N2200	Si	Au	SiO_2_	BGTC	0.28 0.05			[111]
FAPbI_3_	Au	Au	LaF_3_	BGTC	(0.002)	10^4^	1.5	[112]
MAPbI_3_ MAPbI_3_ MAPbI_3_ MAPbI_3_	Si	Au Au Au/MoO_3_ Au/MoO_3_	SiO_2_ SiO_2_ SiO_2_ Al_2_O_3_	BGBC BGTC BGTC BGTC	0.27 2.27 5.28 21.41	10^4^		[113]
MAPbI_3_ Micro plate	Si	Au	SiO_2_	BGBC				[114]
MAPbI_3_	Si	Au/Ti	HfO_2_	BGBC	0.001	10^4^		[115]
MAPbI_3_	Si	Au/Ti	SiO_2_	BGTC	0.02			[116]
MAPbI_3_	PET	ITO/Cu gate	CYTOP	TGBC	1.7			[117]
MAPbCl_3_ MAPbBr_3_ MAPbI_3_ MAPbCl_3_ MAPbBr_3_ MAPbI_3_	Si	Au	SiO_2_	BGTC BGTC BGTC BGBC BGBC BGBC	2.6 3.1 2.9 3.8 3.6 4.7	10^4^ 10^3^ 10^4^ 10^5^ 10^5^ 10^5^	4.4 −4.9 −13.2 1.45 1.15 0.16	[68]
MAPbBr_3_	Si	Au	SiO_2_	BGTC				[118]
MAPbBr_3_/Graphene	Si	Au	SiO_2_	BGBC				[119]
RbCsFAMAPbI_3_	Glass	Au/Cr	Cytop	BGTC	0.5			[69]
CsFAMAPb(BrI)_3_	Si	Au	Y_2_O_2_	BGTC	3.35 (4.02)	10^4^		[120]
Cs_x_(Ma_0.17_FA_0.83_)_1−x_PbBr_0.17_Cl_0.83_	Si		PMMA	TGBC	2.02 (2.39)	10^4^		[76]
MAPbI_3−x_Cl_x_		Au, Al as gate	Cytop	TGBC	1	10^2^		[13]
MAPbI_3−x_Cl_x_	Si	Au, Al as gate	SiO_2_	BGTC	0.29 (0.42)			[121]
MAPbI_3−x_Cl_x_	Si	Au	SiO_2_	BGTC	10			[79]
MAPbI_3−x_Cl_x_	Si	Au	SiO_2_	BGBC	0.018			[122]
FASnI_3_	Si	Au	SiO_2_	BGBC	0.21	10^4^	2.8	[123]
MAPbI_3_	Glass/Mo	IZO	Al_2_O_3_	BGBC	(0.25)		7	[124]
MAPbI_3_ CS_X_MA_1−X_PbI_3_		IZO	Al_2_O_3_	BGBC	0.01 0.04	10^2^ 10^4^		[125]
MAPbI_3_	Glass	Au	ITO	BGTC	0.055	10^4^		[126]
MAPbI_3_	Si	Au	SiO_2_	BGBC	0.22			[127]
MAPbBr_3_	Si	Au	SiO_2_	BGTC				[128]
MAPbI_3_	Si	Au/Ti	SiO_2_	BGBC	0.003	50		[129]
MAPbI_3_	Si	Au	SiO_2_	BGTC	0.00013 (0.086)	10^4^		[130]
MASnI_3_	Si	Au	HfO_2_	BGTC	1.3	10^6^		[131]
MASn(I_x_Br_1−x_)_3_					4.3			
MASn(I_x_Cl_1−x_)_3_					9.5			
MASn(I_x_Cl_y_Br_1−x−y_)_3_					19.6	10^7^	0	

Mobility values given in brackets are Electron mobility; FA—Formamidinium; PMMA—Poly methyl methacrylate.

**Table 3 nanomaterials-12-02396-t003:** The all-inorganic perovskite structured material channel based thin-film field-effect transistor works in the literature after 2012.

Perovskite Channel	Substrate	Source Drain	Insulator	Device Structures	μ_e_/μ_h_ [cm^2^ V^−1^s^−1^]	Ion/Off	V_TH_	Ref
CsPbBr_3_	Si	Au	Pentacene	BGTC				[135]
CsPbBr_3_ Micro plate	Si		SiO_2_	BGTC	0.34 (0.4)	10^3^	−19, 28	[133]
CsPbBr_3_ NC/Poly fluorine	Si	Au	SiO_2_	BGBC	5	10^5^	−5	[136]
CsPbBr_3_	Glass	Au	Ion gel	BGTC				[137]
CsPbI_3_ NC		Au	SiO_2_	BGTC	1.17			[55]
CsPbBr_3_ QDs/MoS_2_	Si	Au/Cr	SiO_2_	BGTC	0.21			[138]
CsPbBr_3_ QDs	Si	Au/Cr	SiO_2_/PMMA	BGTC	328			[139]
CsPbBr_3_ NC	Si	Au	SiO_2_	BGBC	0.001	10^4^		[83]
BaSnO_3_	BaTiO_3_	Ti/Au	BaSnO_3_	TGBC	9.7		−4.5	[71]
CsPbBr_3_ CsPbBr_3_/MEH–PPV	Si	Au/Al	SiO_2_	BGBC	(0.027) 9	10^5^ 10^6^		[140]
CsPbBr_3_ nanoplate	Glass	Au	PMMA	TGBC	0.26	~10^4^		[141]
CsPbI_3_ Nanowire	Si	Ti/Au	SiO_2_	BGTC	3.05	10^4^–10^5^	−10.9	[142]
CsPbBr_3_ Nanowire	Si	Ti/Au	SiO_2_	BGTC	2.17	10^4^–10^5^	−13.9	
CsPbCl_3_ Nanowire	Si	Ti/Au	SiO_2_	BGTC	1.06	10^4^–10^5^	−21.5	
CsPbBr_3_ Ag doped CsPbBr_3_	Si	Au	SiO_2_	BGBC	7 × 10^−7^ 4 × 10^−4^			[143]
CsPbBr_3_	Si	Ag	SiO_2_	BGBC	2.3	10^6^		[82]
CsPbBr_3_	Si	Au/Ti	SiO_2_	BGTC		10^3^		[144]
CsSnI_3_	Si	Au	SiO_2_	BGTC	50	3 × 10^8^	12.5	[145]

**Table 4 nanomaterials-12-02396-t004:** The double and triple perovskite structured material channel based thin film FET studies in the literature after 2012.

Perovskite Channel	Substrate	Source Drain	Insulator	Device Structures	μ_e_/μ_h_ [cm^2^ V^−1^s^−1^]	Ion/Off	V_TH_	Ref
(BA)_2_PbI_4_ (BA)_2_(MA)Pb_2_I_4_ (BA)_2_(MA)_2_Pb_3_I_10_	Si	Ag	SiO_2_	BGTC	(0.002) (0.083) (1.25)	10^2^ 10^4^ 10^6^	30 38 90	[149]
P3HT (PEA)_2_PbCl_4_ (PEA)_2_PbBr_4_ (PEA)_2_PbI_4_	ITO	Au		BGTC	0.005 0.05 0.09 0.14		7.27 8.45 7.5 5.19	[150]
(PEA)_2_SnI_4_ (PEA)_2_SnI_4_, 5% CNT (PEA)_2_SnI_4_, 100% CNT (PEA)_2_SnI_4_, 20% CNT	Si	Au	SiO_2_	BGTC	0.63 0.99 1.66 1.25	10^5^ 10^5^ 10^5^ 10^5^	14.5 9.6 1.1 6.4	[151]
(PEA)_2_PbI_4_	Si	Au	SiO_2_	BGTC				[152]
(PEA)_2_SnI_4_	Si	Au	P(VDF-TrFE)	BGTC	0.62	100		[153]
(PEA)_2_CsSn_2_I_7_	Si	Cr/Au	SiO_2_	BGBC	34			[154]
(PEA)_2_SnI_4_ (PEA)_2_SnI_4_ (PEA)_2_SnI_4_ (PEA)_2_SnI_4_ (PEA)_2_SnI_4_ (PEA)_2_SnI_4_	Si	Au Au Au Au Au/MoO_3_ Au/MoO_3_	Cytop NH_3_I-SAM NH_3_I-SAM NH_3_I-SAM Cytop NH_3_I-SAM	BGBC BGBC BGTC BGTC TGTC TGTC	1.1 3.8 7 9.1 8.6 12	10^5^ 10^6^ 10^6^ 10^6^ 10^6^ 10^6^	−34 −30 −28 −26 −23 −22	[90]
(PEA)_2_SnI_4_	Si	Au Ag Al	NH_3_I-SAM	TGBC	(1.1) (1.3) (1.8)	10^3^ 10^4^ 10^4^	53 48 47	[155]
(PEA)_2_SnI_4_ (PEA)_2_SnI_4_	Si	Al/C_60_ Au/MoO_X_	SiO_2_	BGTC	(35.5) 50	10^7^ 10^6^	30.8 −26.8	[91]
(PEA)_2_SnI_4_		Au	Cytop	BGTC	0.4	10^6^	−73	[156]
(PEA)_2_SnI_4_ (4–Tm)_2_SnI_4_	Si	Au	SiO_2_	BGTC	(0.15) 2.32	10^5^ 10^5^	5−30	[157]
(PEA)_2_SnI_4_	ITO	Au	CL-PVP	BGTC	0.33	10^3^	21	[158]
(PEA)_4_BiAgBr_6_	Si	Au	SiO_2_	BGTC	0.000017	100	−23	[159]
(PEA)_2_SnI_4_	Si	Au	SiO_2_	BGTC	4.11	10^6^	8.2	[160]
((BA)_2_(MA)_n−1_PbnI_3n+1_	Si	Au	SiO_2_	BGTC	0.0025			[161]
Cs_2_AgBiBr_6_					0.00015	10^4^		[148]
Cs_2_AgBiBr_6_	Si	Au	SiO_2_	BGBC	0.29		5–20	[147]
(PEA)_2_SnI_4_ 5mol% SNI_4_ doped (PEA)_2_SnI_4_	Si	Au	SiO_2_	BGBC	0.25 0.68	10^4^ 10^5^	−21 10	[162]
(PEA)_2_SnI_4_ PDVT–10 doped (PEA)_2_SnI_4_	Si	Au	SiO_2_	BGBC	0.1 0.46	10^2^ 10^4^		[163]

PVDT-10—Poly{3,6-dithiophen-2-yl-2,5-bis(2-decyltetradecyl) pyrrolo[3,4-c]pyrrole-1,4-dione-alt-thienylenevinylene-2,5-yl}.

**Table 5 nanomaterials-12-02396-t005:** All the reported perovskite-based photo FETs and their performances.

Perovskite Channel	Mobility (cm^2^V^−1^s^−1^)	Spectral Range nm	I_Light_/I_Dak_ Ratio	Photoresponsivity (AW^−1^)	Rise/Fall Time (μs)	Ref
MAPbI_3−X_Cl_X_		350–1200	10^6^	4 × 10^6^	8000	[174]
(PEA)_2_SnI_4_	0.62	470–625	100	14.57	50,000	[153]
CsPbBr_3_	0.34	532		110		[133]
(PEA)_2_SnI_4_	4.2	550	10^5^		4.8/1.2 × 10^−6^	[175]
MAPbI_3_/C8BTBT	2–6	473	10^3^	30		[19]
MAPbI_3_ MAPbI_3−X_Cl_X_	1.24 1.01	400–800		320	8 × 10^−6^	[97]
MAPbI_3_	1.7			14		[111]

**Table 6 nanomaterials-12-02396-t006:** The works of Perovskite thin-film FET-based Photo detectors in the literature.

Channel Material	Absorber Material	Photo Responsivity AW^−1^	Absorbance Spectral Range nm	Intensity Wcm^−2^	Detectivity (Jones)	Ref
CsPbBr_3_Single crystal	CsPbBr_3_ Single crystal	2	527–538	3.1 × 10^−5^		[181]
CsPbBr_3_	PbS	4.5 × 10^5^	400–1500		7 × 10^13^	[182]
BA_2_PbI_4_	BA_2_PbI_4_	10^4^	532		4 × 10^10^	[183]
FASnI_3_	FASnI_3_	2.8 × 10^3^	300–1000	37 × 10^−5^	1.92 × 10^12^	[105]
(DME)PbBr_4_	(DME)PbBr_4_	132				[184]
CsPbI_3_ Nanorods	C8BTBT	4300	400			[169]
MAPbI_3_/PVDT MAPbI_3_/N2200	MAPbI_3_/PVDT MAPbI_3_/N2200	169 20	450 450		7 × 10^11^ 3 × 10^11^	[111]
(C_4_H_9_NH_3_)_2_(MA)Pb_2_I_7_	(C_4_H_9_NH_3_)_2_(MA)Pb_2_I_7_	1.2 × 10^4^	800–1600			[185]
CsPbBr_3_	CsPbBr_3_	6.48	473		3.86 × 10^11^	[82]
(C_4_H_9_NH_3_)_2_(MA)Pb_2_I_7_	(C_4_H_9_NH_3_)_2_(MA)Pb_2_I_7_	1302		10		[98]
MAPbBr_3_/Graphene	MAPbBr_3_	1017	532	0.66	2 × 10^13^	[118]

**Table 8 nanomaterials-12-02396-t008:** Reported ferroelectric perovskite FETs in the literature with respective ferroelectric and dielectric layers.

Semiconductor	Gate	Ferroelectric Layer	Dielectric	Ref
PZT	LaAlO_3_	LCMO	LCMO	[14]
PZT	TiO_2_	PZT	HfO_2_	[73]
PCBM/MAPbI_3_	La_x_Sr_1__−__x_TiO_3_	PLZT	Graphene	[197]
(PEA)_2_SnI_4_	Au	P(VDF-TrFE)	P(VDF-TrFE)	[155]
Si	La_x_Sr_1__−__x_TiO_3_	BaTiO_3_	La_x_Sr_1__−__x_TiO_3_	[177]
CsPbBr_3_/PMMA	Si	Ag NPs	SiO_2_	[141]
PZT	LaAlO_3_	LSMO	LSMO	[197]

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
