# Peer review of "Review on Perovskite Semiconductor Field–Effect Transistors and Their Applications"

_nanomaterials, 2022, doi:10.3390/nano12142396_

Round 1

Reviewer 1 Report

Abiram et al. focus on the works on perovskite FETs which are summarized into tables based on their structures and electrical properties. This review is well written and adds essential knowledge in the field of material science.

  1. The author should add some latest research work to the introduction. The current references are not new enough to reflect the latest research progress.
  2. Why did the author choose perovskite as the research object? What enlightenment does this review have for other researchers?
  3. This image can be further optimized to improve the image quality.
  4. The structure of the article should be further adjusted. The current structure is confusing.
  5. Several relative papers in nanomaterials are suggested to be cited. (10.1016/j.jhazmat.2021.128062, 10.1021/acsami.1c22035, 10.1016/j.cej.2021.131191).
  6. The resolution of the figure shown should be further improved.
  7. Conclusions should summarise this paper and should not quote the research content in the references.

Author Response

Abiram et al. focus on the works on perovskite FETs which are summarized into tables based on their structures and electrical properties. This review is well written and adds essential knowledge in the field of material science. The author should add some latest research work to the introduction. The current references are not new enough to reflect the latest research progress.

We thank the reviewer for the encouraging comments. We updated the manuscript with recent papers.

  1. Why did the author choose perovskite as the research object? What enlightenment does this review have for other researchers?

We thank the reviewer for the constructive comments. Perovskite is the revolutionary material in the field of photovoltaic for more than a decade. A huge number of perovskites are being studied for the solar cells, but only a few number of perovskites have been studied repeatedly. With excellent optoelectronics properties, perovskites should be explored in the field effect transistor studies.

As we summarized the reported perovskite FET studies along with the electrical properties in the tables, categorized based on the materials, the researches would benefit in getting the basic idea about the performance of existing materials and device structures in the literature. In this review we mainly focus on the applications of the perovskite FETs hence the researches can identify the potential of perovskite FETs to take it to the next level in different applications rather than being on the research level.

  1. This image can be further optimized to improve the image quality.

Thank you and we have worked on improving the quality of the images and the figure is replaced with better ones.

  1. The structure of the article should be further adjusted. The current structure is confusing.

In this manuscript, initially we start with the introduction of perovskite materials and then we shift our attention on the perovskite FETs onto two timelines as only a few studies were carried before 2012. After 2012, number of studies is increased so we categorize them into 3 types based on different classes of perovskite materials. After that the main focus of the study, applications of perovskites FETs are analyzed in 5 different categories.  

  1. Several relative papers in nanomaterials are suggested to be cited. (10.1016/j.jhazmat.2021.128062, 10.1021/acsami.1c22035, 10.1016/j.cej.2021.131191).

Thank you for the comment and we have added more references to the article.

  1. The resolution of the figure shown should be further improved.

Thank you and the figure is replaced with an improved one

  1. Conclusions should summarise this paper and should not quote the research content in the references.

Thank you for this perceptive comment. We have changed the conclusion accordingly

Reviewer 2 Report

Overall, the author summarizes 199 papers with the necessary information on the perovskite FETs and introduces the fundamentals of perovskite materials, their applications, and challenges straightforwardly. This review is very interesting and should be published in nanomaterials after some minor updates are made. The reviewer has the following question:

In section five, the author mentions the challenges of perovskite in the FET applications due to the hysteresis effect. Can the author briefly summarize the challenge in perovskite materials for memory applications? For example, perovskite ferroelectric's low coercive field value will strongly limit its thickness for memory applications.

Author Response

Overall, the author summarizes 199 papers with the necessary information on the perovskite FETs and introduces the fundamentals of perovskite materials, their applications, and challenges straightforwardly. This review is very interesting and should be published in nanomaterials after some minor updates are made.

The reviewer has the following question: In section five, the author mentions the challenges of perovskite in the FET applications due to the hysteresis effect. Can the author briefly summarize the challenge in perovskite materials for memory applications? For example, perovskite ferroelectric's low coercive field value will strongly limit its thickness for memory applications.

Thank you for the insightful comments and feedback. We have made the minor updates on this to be published in nanomaterials. We have included the challenges of memory applications based on perovskite FETs based on some newly added references.

Round 2

Reviewer 1 Report

After the first run of revisions, the manuscript improved in quality. However, I will recommend it be published until the following comments are addressed.

1.      The conclusion should be the summary of the paper instead of adding Figure 14. Please modify again.

2.      It is suggested that the author briefly explain what enlightenment this work has for other researchers.

Author Response

  1. The conclusion should be the summary of the paper instead of adding Figure 14. Please modify again.

We thank the reviewer for the comment and the conclusion is rewritten

  1. It is suggested that the author briefly explain what enlightenment this work has for other researchers.

We thank the reviewer for the comment. Our primary aim in this review paper is to explore the potential application of the perovskite FETs along with their electronic properties. The reviewer first summarises all the FET works and then we categorised the works according to the applications.

Reviewer 2 Report

This revised version, the authors have revised preciously the manuscript based on all reviewers' comments (point by point) in detail. The revised version overall quality is “excellent”. I recommend right now this version is more contribution and suitable for publication.

Author Response

We thank the reviewer for their time in reviewing and accepting our manuscript.